**Perspective**

# Elucidating reactive sugar-intermediates by mass spectrometry
Chun-Wei Chang [1,2], Dana Wehner [1,2], Gurpur Rakesh D. Prabhu [1,2], Eunjin Moon [1,2], Marc Safferthal [1,2], Leïla Bechtella [3], Nicklas Österlund [1,2], Gaël M. Vos [1,2] & Kevin Pagel [1,2] ✉

The stereoselective introduction of glycosidic bonds is one of the greatest challenges in carbohydrate chemistry. A key aspect of controlling glycan synthesis is the glycosylation reaction in which the glycosidic linkages are formed. The outcome is governed by a reactive sugar intermediate - the glycosyl cation. Glycosyl cations are highly unstable and short-lived, making them difficult to study using established analytical tools. However, mass-spectrometry-based techniques are perfectly suited to unravel the structure of glycosyl cations in the gas phase. The main approach involves isolating the reactive intermediate, free from external influences such as solvents and promoters. Isolation of the cations allows examining their structure by integrating orthogonal spectrometric and spectroscopic technologies. In this perspective, recent achievements in gas-phase research on glycosyl cations are highlighted. It provides an overview of the spectroscopic techniques used to probe the glycosyl cations and methods for interpreting their spectra. The connections between gas-phase data and mechanisms in solution synthesis are explored, given that glycosylation reactions are typically performed in solution.

Carbohydrates are ubiquitous and involved in many biological processes[1]. Glycoconjugates, which are carbohydrates linked to entities such as peptides, lipids, or proteins, mediate a plethora of physiological functions. They are involved in processes including cellular communication and immune response and are used as entry receptors by viruses[2,3]. Oligosaccharides have a large structural diversity due to their non-template driven biosynthesis. They can differ in their monosaccharide composition, connectivities, and linkage configurations. Their structural complexity increases exponentially with size and poses a long-standing challenge for analytical and synthetic chemists, which aim to obtain isomerically pure glycans with defined regio- and stereochemistry for further functional studies[4,5].

Synthetic oligosaccharides have been pivotal for the development of glycoconjugate vaccines, microarrays to understand pathogen interactions with glycans, and probes for biological research and drug discovery[6]. At the heart of carbohydrate synthesis is a coupling step between sugar building blocks - the glycosylation reaction. Glycosylation links a glycosyl donor with a nucleophile, which can be a variety of biomolecules, such as peptides, proteins, or glyco-alcohols[2,7,8]. Because the reaction mechanism spans the spectrum between $S_N1$ and $S_N2$ mechanisms, two stereoisomers can be formed: α- and β-glycosides[9]. Controlling the stereoselectivity of the glycosylation reaction presents a significant bottleneck in carbohydrate chemistry[10]. Generally, the stereoselective outcome is optimized through extensive empirical tests, but many inconsistent results are observed due to the poor understanding of the reaction mechanism[11,12].

Gaining insight into the glycosylation processes is essential for controlling glycoside formation. Particularly, the outcome of glycosylation is largely determined by a highly reactive sugar intermediate — the glycosyl cation. This intermediate structure plays a key role in guiding the nucleophile approach. This, in turn, shapes the α- or β-configuration of the resulting glycosidic bond.

The goal of mechanistic studies is to understand the behavior of glycosyl cations. This involves presenting a clear and step-by-step picture of how the intermediate participates in the reaction. However, the exact structure of glycosyl cations has remained unclear thus far. This lack of fundamental understanding is largely a result of their extremely short-lived nature. Their lifetimes are estimated to be just a few picoseconds in solution[13,14]. In addition, glycosylation is a complex chemical process. It is influenced by various factors, including solvents and acids. These large number of interconnecting factors directly impact the structure and stability of the intermediate. Due to these challenges, it is difficult to analyze glycosyl cations using conventional spectroscopic techniques[15].

Mass spectrometry (MS) has significantly improved our understanding of the glycosyl cation and their role in glycosylation reaction mechanisms. The primary advantage of MS is the cleanroom environment (i.e., high vacuum), which allows to stabilize glycosyl cations for seconds and

[1]Freie Universität Berlin, Institute of Chemistry and Biochemistry, Berlin, Germany. [2]Fritz Haber Institute of the Max Planck Society, Berlin, Germany. [3]Université Paris-Saclay, Univ Evry, CY Cergy Paris Université, CNRS, LAMBE, 91025 Evry-Courcouronnes, France. ✉e-mail: kevin.pagel@fu-berlin.de

detect them without interference from solvents or other reactants[16]. Integrating orthogonal gas-phase technologies has enabled an in-depth structural analysis of the ions[17,18]. For example, the combination of ion mobility and MS reveals information on the shape and size of an ion[19–23]. The target ion can also be probed by gas-phase infrared (IR) spectroscopy to investigate the vibrational patterns of functional groups and elucidate its three-dimensional structure[18,24].

This perspective highlights recent MS-based efforts to unravel the structure of glycosyl cations. It focuses on methodologies involving gas-phase IR action spectroscopy and ion mobility-mass spectrometry (IM-MS). MS-based techniques for structural characterization are presented, the setups of IR spectrometers are explained, and their capabilities are discussed. Furthermore, it is highlighted how spectroscopic and spectrometric data are interpreted through computational calculations. Finally, an examination is provided on how gas-phase data can help our understanding of the mechanism of glycan synthesis in solution and observed correlations as well as limitations are discussed.

## A historical resume of glycosylation reaction research

In the late 19[th] century, Emil Fischer reported the first successful glycosylation reaction (Fig. 1a)[25]. Since then, glycosylation research has undergone extensive development, though its fundamental principles have remained largely unchanged. The key step is the formation of glycosyl cations, which determines the stereochemistry of the glycosidic bond at the anomeric carbon. Chemical synthesis has been employed to study the mechanism of the glycosylation reaction by capturing the reactive intermediate. The utilized methods include the use of [13]C kinetic isotope effects[26,27], cation clocks[28,29], and trapping experiments[30–32]. In these experiments, the behavior of intermediates during the reaction is investigated by analyzing the structure of isolated products. From an energy perspective, the $S_N2$ mechanism forms covalent intermediates that are thermodynamically stable and located at an energy minimum[33]. These covalent species were first discovered using low-temperature NMR[34]. Crich and coworkers provided clear structural evidence of α-glycosyl triflates which has led to a resurgence of NMR-based approaches to study glycosylation intermediates[35,36]. The presence of the α-glycosyl triflate explains β-selectivity through a stereo-inversion ($S_N2$) mechanism. The $S_N2$ pathway was further refined by in-depth analysis using exchange NMR spectroscopy (EXSY)[37,38]. Introduction of chemical exchange saturation transfer (CEST) NMR further detected low-abundant species[39,40]. Recently, variable temperature NMR (vt-NMR) was applied to study the glycosylation reaction in thioglycoside activation systems[41–43]. The data validated counter-ion exchange through the participation of byproducts and demonstrated a dynamic balance of triflate and halide species in triflate promoted system.

The $S_N2$-glycosylation has been well-studied using NMR[34]. However, the behavior of glycosyl cations that are responsible for the $S_N1$-product remains poorly understood. In the $S_N1$ pathway, glycosyl cations are transient and present in an ionic form. These short-lived cationic species can be easily trapped by solvents and promoters in solution, resulting in a highly dynamic picture[10–12] (Fig. 1b). The intermediate transformation is sensitive to environmental factors and sugar conformation. Even slight changes in heat can push the equilibrium towards more unstable states[44–47]. Because of these challenges, it is difficult to directly measure glycosyl cations using traditional spectroscopic techniques. This limitation has been emphasized since 2011 by Yoshida et al.[48,49].

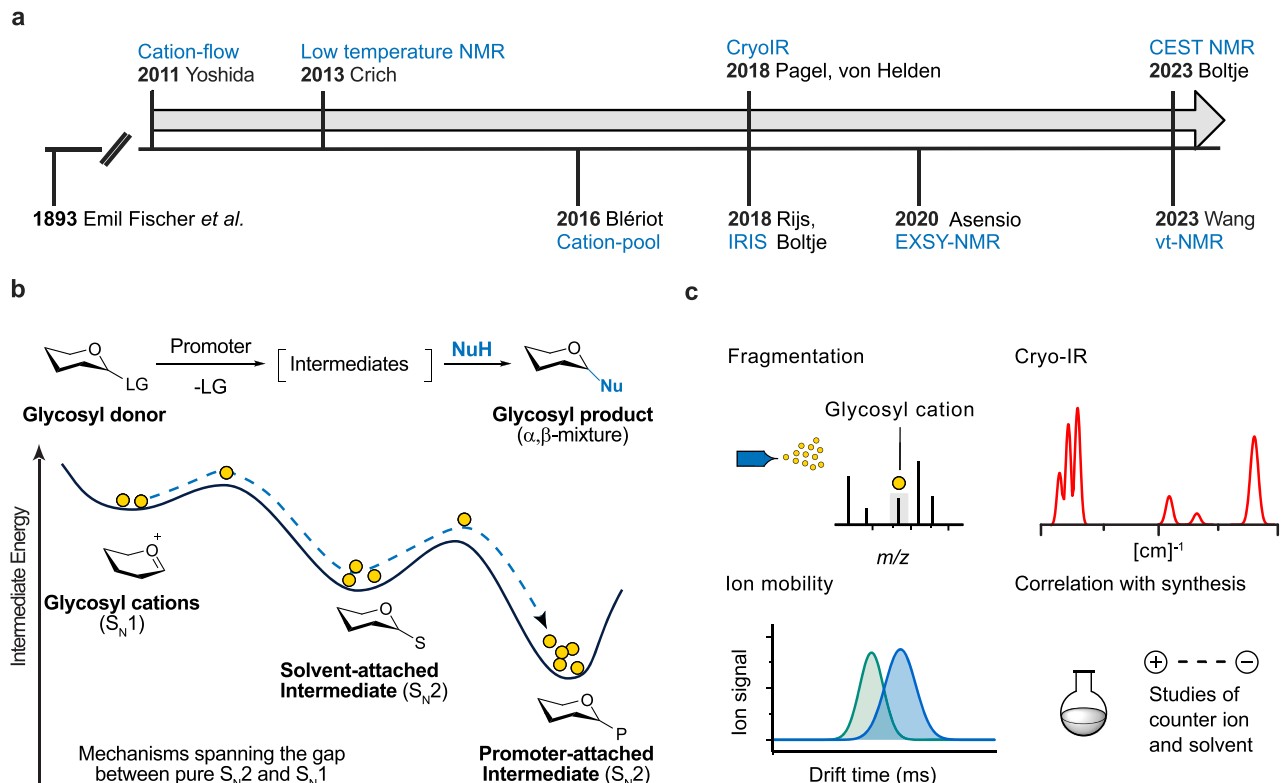

**Fig. 1 | The evolution of glycosylation reaction research: from 19[th] century innovations to modern scientific advances. a** The timeline of the discovery of the glycosyl intermediates. The pioneering work of Emil Fischer reported the first successful glycosylation reaction. Low-temperature NMR has been introduced to elucidate the covalent intermediate. The MS-based technology detects glycosyl cations in the gas phase. Structural data of intermediates improves the understanding of the glycosidic bond formation. **b** Energy landscape of glycosyl intermediates. Glycosyl cations have notably short lifetimes and can be easily trapped by solvents and promoters, creating a highly dynamic mechanistic picture. NuH nucleophile, S solvent, P promoter, LG leaving group. **c** Exploring the structure of glycosyl cations in the gas phase: Infrared spectroscopy shows vibrational patterns. Ion mobility-mass spectrometry provides insights on the molecular shape. Studies on counter ion and solvent effects is key to connect gas-phase data to solution behavior.

A clear structural definition of glycosyl cations is essential for a better understanding of the glycosylation reaction. According to current understanding[35,36], glycosyl cations are positively charged molecules that form under acidic conditions. They originate from glycosyl donors when activated by an acid-based catalyst. Once formed, these cations are surrounded by counter-ions and solvent molecules. Blériot introduced the cation-pool method to capture this elusive intermediate[50–52]. The structure is examined by NMR in super acid, where the glycosyl cations are weakly coordinated with the $SbF_6^-$ anions. Recently, Thibaudeau and coworkers have refined the superacid methodology to capture naked sugar cations without protective groups, offering deeper insights into the transition state of enzymatic reactions[53].

A recent and promising alternative to study glycosyl cations is the integration of MS with vibrational spectroscopy[54]. To generate the glycosyl cations in the gas phase, electrospray generated ions of precursor molecules with a labile leaving group are thermally activated in the initial low-vacuum region (e.g. ion-guide or ion-funnel regions; see below) of the mass spectrometer, where they can collide with neutral atmospheric gas molecules to generate glycosyl cations by in-source collision-induced dissociation (CID). The transferred energy leads to bond dissociation at the anomeric carbon to form glycosyl cations[54–59]. Alternatively, to increase selectivity, the precursor ions may be isolated based on $m/z$ and fragmented in a more controlled atmosphere of neutral gas molecules by CID (in an ion trap or collision cell; see below)[60,61]. The stability of the glycosyl cation is greatly increased because the ion is isolated in a high-vacuum environment, free from solvent and promoter that would interact otherwise. Hyphenation with other gas-phase analysis techniques then allows further structural characterization. So far, efforts have focused on employing orthogonal gas-phase techniques such as IR spectroscopy and ion mobility spectrometry (IMS) combined with computational calculations to interpret the outcome (Fig. 1c)[17,18,62].

## Ion mobility spectrometry

IMS is a gas-phase technique that separates ions according to their size and shape, rather than mass, making it particularly powerful for resolving isomeric ions which cannot be separated by MS alone. Separation in IMS is achieved as analyte ions display differing mobilities in a buffer gas when subjected to an external electric field. In this process, the electric field ($E$) exerts a force on the ion which, in the steady-state situation, is counterbalanced by a frictional force caused by collisions with the buffer gas. This balance results in a constant ion drift velocity $v_d$. The mobility ($K$) is the proportionality constant that describes to which velocity the ion is slowed by friction (Eq. (1)).

$$K = \frac{v_d}{E} \tag{1}$$

The relationship in Eq. (1) only holds in the so-called low-field limit of the electric field, as the relationship between ion velocity and electric field becomes non-linear at higher electric fields due to additional ion heating. Under low-field conditions, $K$ can be further expressed in terms of specific physical parameters of the ion and buffer gas using the Mason-Schamp equation (Eq. (2)).

$$K = \frac{3}{16} \frac{ze}{N} \sqrt{\frac{2\pi}{\mu k_B T}} \frac{1}{\Omega} \tag{2}$$

The Mason-Schamp equation describes the mobility in terms of several properties of the analyte ion and buffer gas, including the reduced mass of the analyte ion-buffer gas pair ($\mu$), the charge of the analyte ion ($z$), the buffer gas temperature ($T$), and the gas number density of the buffer gas ($N$), along with the collisional cross section (CCS, $\Omega$) of the analyte ion-buffer gas pair. The CCS is a shape-factor which can be thought of the rotationally averaged collision surface of an ion that interacts with the buffer gas (Fig. 2a). It is an intrinsic, instrument-independent property which carries useful structural

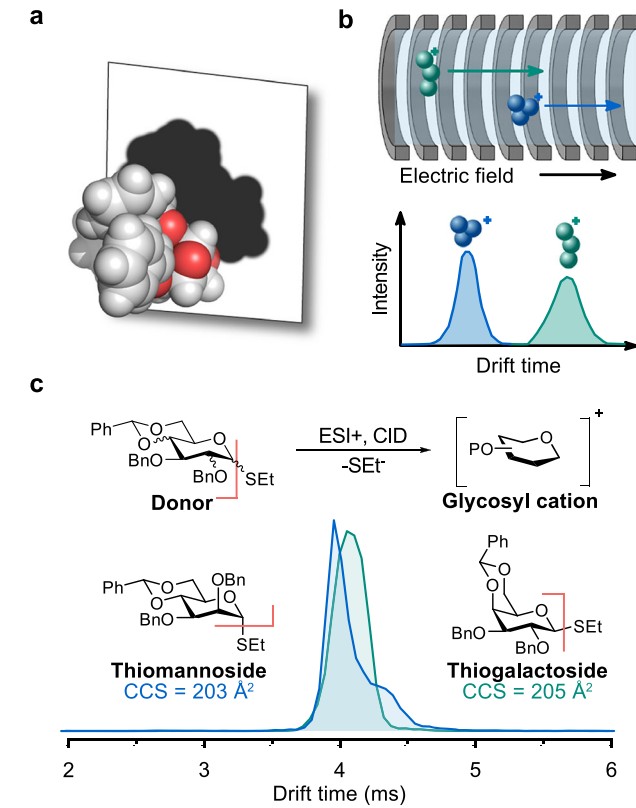

**Fig. 2 | Ion mobility spectrometry (IMS). a** Collision-cross section represented as a 2D projection of a molecule. **b** Scheme of a drift tube IMS cell and depiction of an arrival time distribution. **c** Glycosyl cation measured using traveling-wave IMS. The difference between galactosyl and mannosyl cations lies in the C2 stereochemistry, causing distinct arrival time distributions and CCSs. SEt thioethyl; ESI electrospray ionization, CID collision induced dissociation, CCS collision cross section. Adapted from https://doi.org/10.1038/s44160-024-00619-0[58], under the terms of the CC BY 4.0 license, https://creativecommons.org/licenses/by/4.0/.

information about the analyte ion. The CCS has the dimensionality of a surface and is typically reported in $Å^2$.

The most intuitive way to perform IMS is by so-called drift-tube (DT) IMS where the drift time ($t_d$) through an IMS cell of well-defined length (l) is measured upon applying a weak and uniform electric field (Fig. 2b). As DTIMS is performed at the low-field limit, Eq. (1) can be rearranged, which provides for a direct way of determining K (Eq. (3)).

$$K = \frac{l}{t_d E} \tag{3}$$

Usually, the mobility is transformed into the reduced ion mobility $K_0$ which is normalized to standard temperature and pressure ($T_0 = 273.16$ K and $p_o = 760$ Torr, Eq. (4)).

$$K_0 = K \frac{T_0}{T} \frac{p}{p_0} \tag{4}$$

The CCS can in this way be calculated directly from DTIMS data using the Mason–Schamp equation, when $K_0$ has been determined.

Besides DTIMS, multiple other methods to separate ions according to their mobility have been developed, which have been comprehensively discussed by Dodds and Baker in a recent review[63]. A very common IMS technique employed in modern commercial instruments is traveling-wave IMS (TWIMS), where shorter drift regions with alternating fields are combined. Mobility separation is achieved in a pulsating electric field where

ions are both accelerated and decelerated by each traveling wave. The clear advantage is that TWIMS achieves higher resolving power with shorter IMS cells compared to DTIMS. Recently, new approaches based on TWIMS have also been developed which have further improved the resolving power[64], these include structures for lossless ion manipulation (SLIM)[65] and cyclic IMS[66]. The main drawback of these techniques is that they employ non-uniform electric fields, which are often also operated outside the low-field limit. This means that direct determination of $K_0$, and hence CCS, is not possible. CCS values must instead be obtained by external calibration using ion standards of known CCS.

CCS values, either calculated from the first principle using DTIMS data or obtained by calibration, can often be used to distinguish isomeric structures if their size difference is sufficiently large to be resolved by the IMS technique employed. The CCS in itself is, however, not unique to a single structure, and matching experimental CCS values to structures often requires computational tools. Various methods have been developed to calculate CCS values computationally from structural models. Some examples of software used for these calculations include MOBCAL[67–69], IMoS[70–73] and HPCCS[74]. Density functional theory (DFT) calculations can in turn provide optimized 3D structures which can be used as input for the CCS calculations.

Following the mobility of an ion while modulating its gas-phase activation enables for monitoring structural changes in the ion inside the instrument. IM-MS instruments allowing for CID before the mobility cell, such as the Waters Synapt G2-S TWIMS instrument, are well suited to study glycosyl cations. Recently, this method was used to investigate the structure of glycosyl cations from various benzylidene-protected monosaccharides (Fig. 2c)[58]. CID activation efficiently cleaves the thioethyl group (SEt) of the glycosyl donor through a heterolytic process, forming a glycosyl cation. Once the glycosyl cations are formed, they are isolated using *m/z* selection and investigated using traveling-wave IMS. The obtained CCS can provide valuable structural information; for example, the difference between galactosyl and mannosyl cations lies in the stereochemistry at the C2 position. This structural variation leads to distinct ion mobility distributions, which appear as unique peaks in the mobilogram and different CCS values obtained through calibration[75]. The CCS values obtained from small oligosaccharide cations can also be used to identify molecular features in larger oligosaccharides, as identical glycan fragments can be generated from termini of larger structures[19,76].

## Gas-phase IR ion spectroscopy

The vibrational spectroscopic analysis of gaseous ions was first introduced in the 1970s[77–79]. However, Coulomb repulsions between ions limit the analyte density in the gas phase, which is too low for direct absorption spectroscopy. To address this problem, action spectroscopy was developed, where the photochemical response of ions is recorded[80]. Instead of a high analyte concentration, a high photon density is required[81–83]. Early instruments developed for action spectroscopy implemented lasers with a narrow wavelength range (e.g. $CO_2$ laser, 900–1100 cm$^{-1}$) alongside Fourier transform ion cyclotron resonance (FT-ICR) MS[81,84,85]. Gas-phase ion spectroscopy took off at the turn of the millennium due to the emergence of high photon flux lasers such as widely tunable free-electron lasers (FEL)[82,83,86–91] and, optical parametric oscillator/amplifier (OPO/OPA) systems[92–97]. Currently, the IR-FEL facilities in Berlin (FHI-FEL)[54–58], Nijmegen (FELIX)[40,60,98], and Orsay (CLIO-FEL)[99] have integrated mass spectrometers for the direct structural analysis of intact molecular ions or short-lived fragments through spectroscopy. These IR techniques broadened the traditional glycomics toolkit and offered remarkable insights into the structures of various biomolecules.

The action IR spectrum can be recorded by detecting, for example, fragmented ions generated through infrared multiple photon dissociation (IRMPD)[94,100]. In IRMPD, an ion absorbs multiple infrared photons (Fig. 3a), the energy from these photons is then spread out across the molecule through intramolecular vibrational energy redistribution (IVR). IVR allows the molecule to dissipate and store the energy internally, leading

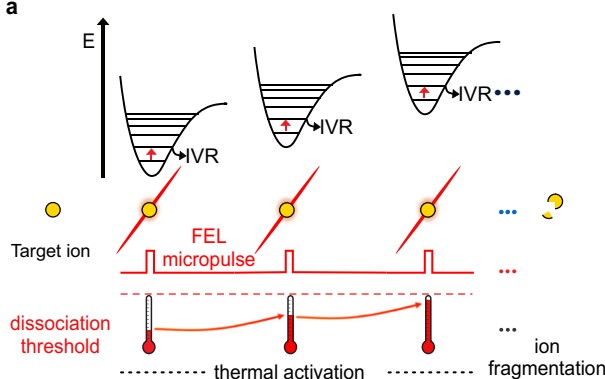

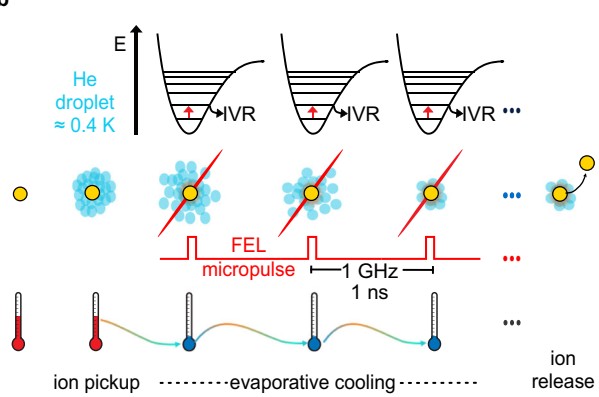

**Fig. 3 | Gas-phase infrared ion spectroscopy scheme of action. a** Infrared multiple photon dissociation (IRMPD); **b** cryogenic helium droplet ion spectroscopy. FEL free-electron laser, IVR intramolecular vibrational redistribution.

to molecular heating. After several cycles of energy absorption and redistribution, the molecule accumulates enough energy to break its chemical bonds. The fragments produced from this process are monitored using MS and plotted as a function of the tunable laser wavelength to yield an IR spectrum[80].

An alternative strategy to perform action spectroscopy is messenger tagging[101]. This technique involves the attachment of a "messenger tag" atom or molecule to the target ion. The messenger tag is attached through weak van-der-Waals interactions by cooling the ion to low temperature (10–60 K). When the ion absorbs a resonant photon, it releases the messenger tag, leading to a shift in *m/z* values of tagged and untagged ions. By tracking the ratio of untagged/tagged ions with varying laser wavelengths, an IR spectrum can be constructed[102,103]. Since messenger tagging is a single photon event, at least in an ideal case, the experiments can be performed with low-intensity tunable table-top lasers such as OPO/OPA systems.

Building on this idea, von Helden and colleagues developed an instrument to perform spectroscopy in superfluid helium droplets and measure cryogenic IR spectra (Fig. 3b)[104,105]. Ions are captured in superfluid helium droplets, rapidly cooling them to sub-kelvin temperatures (around 0.4 K). Once the ions are embedded in the helium droplets, they are irradiated by a FEL beam, absorbing resonant photons leading to excitation. The thermal energy of the ion is quickly transferred to the helium droplet, causing a part of the helium atoms to evaporate, which reestablishes sub-kelvin temperatures. This process is repeated over several cycles of photon absorption and energy dissipation leading to ion release. The intact ion is detected by a time-of-flight (TOF) mass analyzer. The IR spectrum is constructed by plotting the ion signal as a function of the tunable laser wavelength.

Cryogenic technology offers significant advantages to room-temperature IR spectroscopy. Cooling the ions thermally depopulates the number of quantum states. Since excitation is mostly performed from the

vibrational ground state, spectral congestion is reduced. A comparison of cryogenic IR spectra and room-temperature IRMPD spectra demonstrates the effects of cooling (Fig. 4)[106]. In the upper IRMPD spectrum, two broad features in the 1400–1800 cm⁻¹ region and one band in the 1000–1400 cm⁻¹ can be observed. In contrast, the lower cryogenic IR spectrum reveals more detailed information as the vibrational bands are better resolved.

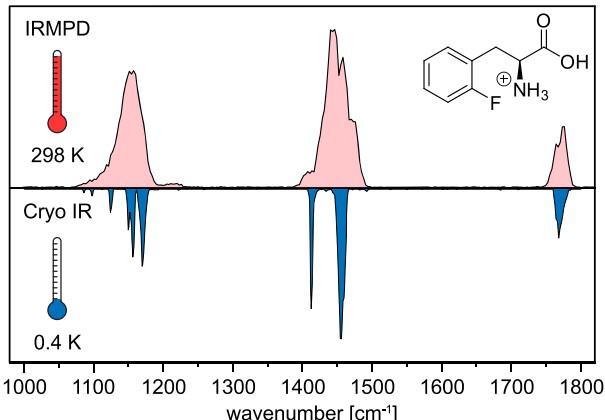

**Fig. 4 | Comparison of experimental IR spectra obtained from room-temperature IRMPD spectroscopy (red) and cryogenic infrared spectroscopy in helium droplets (0.4 K, blue).** The example here is protonated ortho-fluorophenylalanine. Adapted from report[106].

## Brief overview of IR-MS instruments

Hyphenation of IR spectroscopy with MS is often performed on instrumental setups that are based on modified commercial mass spectrometers. Detailed description of IRMPD instruments used for the analysis of glycosyl cations are provided by the developers in Nijmegen[100] and Orsay[94], but a brief overview of the setups is provided here.

The IRMPD instrument at FELIX in Nijmegen is based on a commercial Bruker amaZon speed ETD with an optical access installed at the top and bottom of the ring electrode. In the setup, the FEL beam irradiates the ion cloud trapped in a Paul-type 3D ion trap (Fig. 5a). The setup in Orsay implements a commercial Bruker Apex Qe 7T FT-ICR MS equipped with a collision cell. Irradiation of the analyte is achieved by the CLIO-FEL (for 900–1800 cm⁻¹) or an OPO/OPA (for 3200–3700 cm⁻¹) for IRMPD spectroscopy (Fig. 5b)[94]. The laser beam is directed towards the ICR cell through an IR transparent window at the back of the instrument. Notably, spectra recorded by room temperature IRMPD often exhibit significant peak broadening and red shifting of peak positions. Oomens and co-workers recently commissioned another multi-modal platform that combines liquid chromatography, trapped-IMS (Bruker TIMS 0.5), and IRMPD spectroscopy in a commercial ultra-high-resolution FT-ICR mass spectrometer (Bruker SolariX XR)[107].

An instrument for cryogenic gas-phase ion spectroscopy was developed by von Helden and colleagues and used to study transient glycosyl cations (Fig. 5c)[104,105]. The front end of the setup is based on a Waters Q-TOF Ultima that handles ionization, fragmentation, and selection of ions. Using in-source CID, the precursors are activated forming glycosyl cation intermediates. The cations are transferred to a hexapole cryogenic trap (90 K) for storage. Superfluid helium clusters, adjustable in size from 10⁴ to 10⁶ helium atoms depending on the temperature of the nozzle (15-25 K), are generated

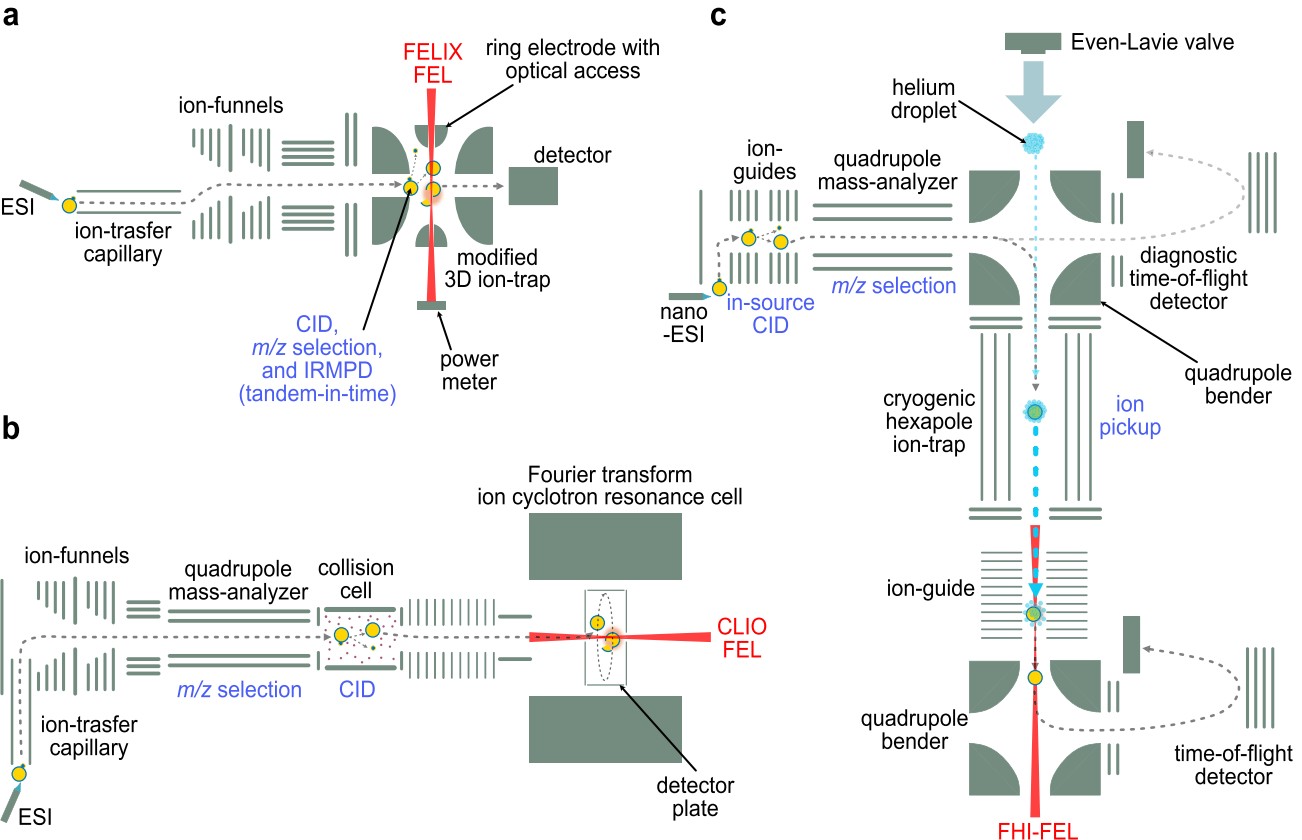

**Fig. 5 | Instrumentation for gas-phase ion spectroscopy. a** IRMPD instrument in Nijmegen, **b** IRMPD instrument in Orsay, **c** helium droplet instrument in Berlin, CID collision-induced dissociation, CLIO Infrared Laser Center of Orsay, ESI electrospray ionization, FEL free-electron laser, FELIX Free Electron Lasers for Infrared Experiments, FHI Fritz Haber Institute.

by a pulsed Even-Lavie valve[108,109]. These droplets – traveling at a speed of about 500 m/s –can capture the ions stored in the trap and carry forward, overcoming the trapping barrier through their kinetic energy. Superfluid helium clusters are transparent to IR light and interact only weakly with the embedded ions. The embedded ions are rapidly cooled to sub-kelvin temperatures (around 0.4 K). The embedded ions are guided towards a time-of-flight (TOF) detector following a pathway that overlaps with the FEL beamline. After multiple photon absorption and energy transfer events, the helium droplets evaporate and the bare ions are detected using the TOF[110]. Several accounts from the instrument developers offer detailed descriptions of the instrument design and operation[54,105].

The first commercial IR-IM-MS instrument, the PhotoSynapt, was developed by MS Vision in collaboration with Rijs and co-workers as an upgrade to the Waters Synapt G2 and G2-S systems[111]. The upgrade introduces an IRMPD chamber for spectroscopy between the TWIMS cell and the TOF[112].

Together with Isospec Analytics, a new version of the PhotoSynapt, the Cold Photo Synapt is currently in development. It allows to perform messenger tagging spectroscopy using a multistage cryogenic trap design capable of parallel ion selection[113]. It is expected that the emergence of commercial IR mass spectrometers and the increased availability of tunable table-top laser technology will lead to an increased implementation of gas-phase ion spectroscopy.

## Computational approaches for spectral analysis

Computational analysis has proven its value in spectroscopy to support and interpret experimental data by enabling reference-free structural identification[114,115]. This identification method is indispensable in glycosyl cation research due to the very short-lived nature of glycosyl cations and the absence of reference spectra. The ability to obtain IR spectra of glycosyl cations resulted in the demand for workflows to generate reference spectra of the various potential reaction intermediate structures. Several labs have developed their own specialized workflow[58,114] (Supplementary Table S1), yet the general steps are similar (Fig. 6).

Initial candidates are chosen by chemical intuition, knowledge of potential structures, or search in molecule databases[116–120]. After candidate generation, a conformational search has to be performed. This is because the vast conformational space of the molecule makes it difficult to identify the lowest-energy structure without screening. Commonly applied tools for conformational search are the CREST (Conformer-Rotamer Ensemble Sampling Tool) program[121,122], the genetic algorithm FAFOOM[123] (Flexible Algorithm for Optimization of Molecules) and force fields. The choice of method for the conformational screening determines the accuracy of the resulting conformer. If the conformational search was performed with DFT-based methods or with sufficiently accurate methods, e.g., GFN2-xTB[124] in CREST, usually only one subsequent geometry optimization and frequency calculation on a higher level of DFT theory with Gaussian 16[125] and/or ORCA[126] is required. If a faster but less accurate conformational search (e.g. GFN-FF) is employed, additional preliminary optimization on a lower level of theory is required for a more reliable ranking of the individual conformers.

A two-step DFT optimization is commonly applied after using a combination of a distance geometry algorithm and classical force fields like the MMFF94[127–131], for the conformer search. An example of the two-step DFT procedure involves the use of the semi-empirical PM6 level[132] for preliminary optimization, followed by optimization and frequency calculation using B3LYP[133,134]/6-311 + G(d,p)[135–137] after a second filtering of conformers by relative energy and similarity. To validate and further

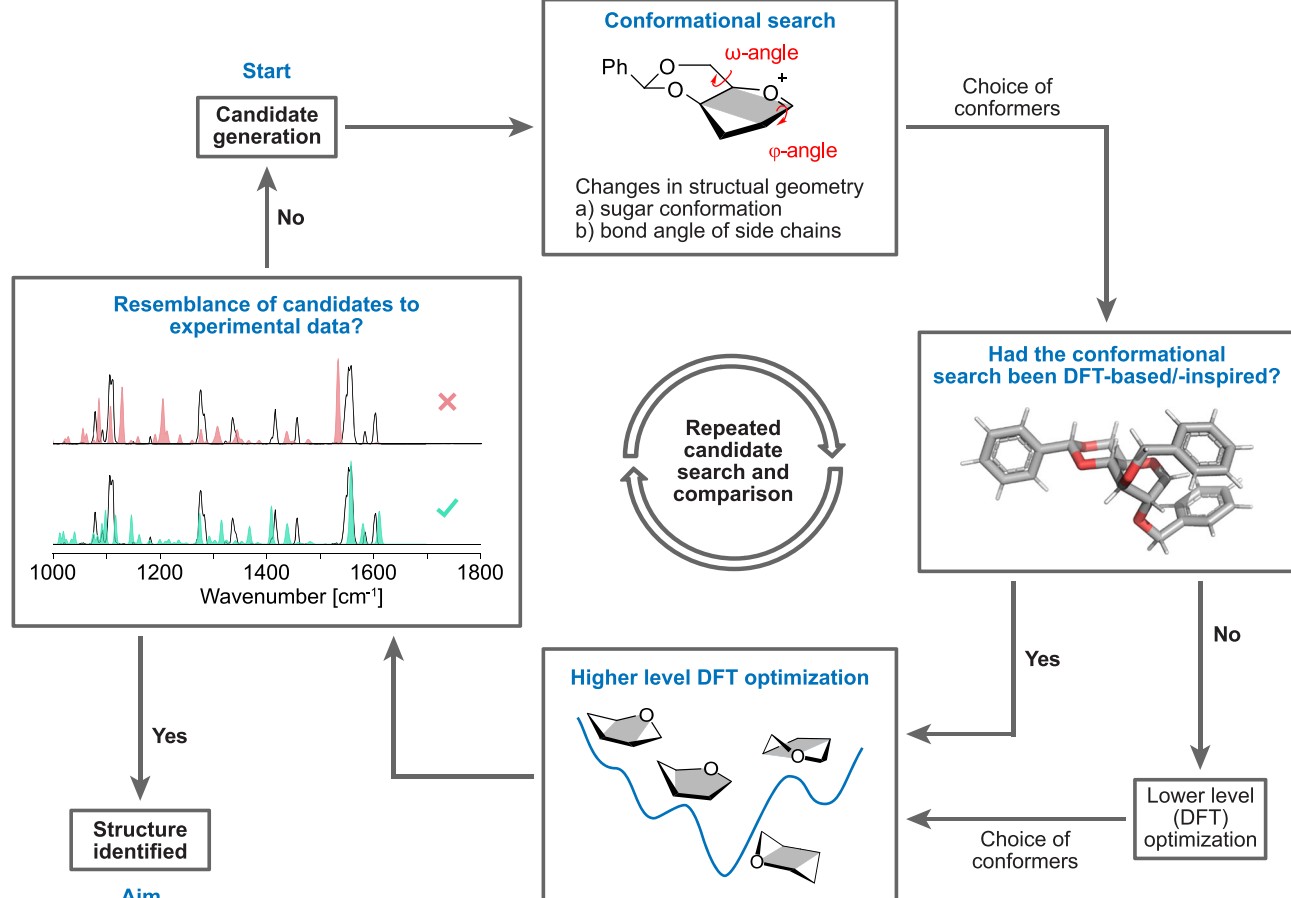

**Fig. 6 | General computational workflow to explore the conformational space of glycosyl cations and identify experimentally measured intermediates.** This workflow can be applied to various molecules and candidates assuming different solvents and counter ions to enable correspondence between gas-phase and solution-phase data.

improve the derived data, the computational investigation can include calculations on two higher levels of theory, with the higher level ensuring the most accurate energy calculation based on the previous DFT results[138]. Moreover, screening with CREST can also be followed by a two-step DFT calculation procedure, if the accuracy during the conformational search is not high enough and justifies the cost of an additional quantum chemical calculation[139].

Based on the calculated frequencies and relative energies, IR spectra can be compared with the experimental spectra. For this comparison, the lowest-energy conformer is generally chosen, and additional processing of the frequencies (scaling, Gaussian-broadening) and the energies (thermal corrections) can be performed. Several scoring tools[140–144] have been developed to assess the similarity between computed and experimental spectra, however, expert interpretation is currently still required.

The capability of computational calculation to assist the identification of glycosyl cations is demonstrated in Fig. 7A. The example shown here are benzylidene-protected sugar building blocks, which are widely applied for constructing β-mannosyl linkage[26]. The cold-IR spectrum of the intermediate was obtained via fragmentation of a benzylidene-protected precursor[58]. By applying the computational workflow, IR spectra for the potential candidates were generated and used for identification. The simulated IR spectra of anhydro cations matched well with the experimental spectrum. The computed spectrum is especially useful in the identification of vibrational patterns of key functional groups. In this example, the PhCHO⁺ moiety of anhydro cations provided the most diagnostic pattern and matched poorly with calculated anomeric oxonium species. The stretching at 1557 cm$^{-1}$ corresponds to the C=O$^+$ stretch and the signals at 1610 and 1579 cm$^{-1}$ correspond to C=C stretches in the phenyl ring. The peaks at 1437 and 1408 cm$^{-1}$ relate to bending in the CHO$^+$ and Ph parts. Besides, the peaks below 1200 cm$^{-1}$ reflect C-O and C-C stretching modes matching a ring pucker in a $^1C_4$ conformation. The cryogenic IR data suggests an intermediate rearrangement in the gas phase (Fig. 7b). The

benzylidene acetal ring opens, creating a zwitterionic species. After that, a bond between O6 and the anomeric carbon is formed, resulting in an anhydro cation. This intermediate rearrangement was the lowest in energy and generated a ring-flip chair form ($^1C_4$). The anhydro cation is thermodynamically stable because the positive charge is delocalized in the benzylium (PhCHO⁺) group.

## Analysis of glycosyl cations using gas-phase IR spectroscopy

The three-dimensional shape of carbohydrates directly influences their function in living organisms. The structure of a carbohydrate can, for example, differ in the configuration at the anomeric carbon (C1) where the orientation of the glycosidic bond leads to two possible forms, 1,2-*cis* glycosides and 1,2-*trans* glycosides (Fig. 8a). Both anomers are important and commonly found in nature. The distinction between the two anomers depends on the relative positions of the C1-glycosidic bond and the C2-hydroxyl group in the sugar ring. For example, 1,2-*trans* glycosides are formed when the C1 group and the C2 group are on opposite sides. 1,2-*cis* glycosides, on the other hand, are formed when the C1 group and the C2 group are on the same side. Under this definition, β-glucoside is a 1,2-*trans* glycoside, and α-glucoside is a 1,2-*cis* glycoside. In mannose, the C2-hydroxyl group is in an axial position, which changes the relative orientation. As a result, α-Mannoside contains a 1,2-*trans* linkage while β-mannoside is 1,2-*cis* linkage.

In carbohydrate chemistry, protecting groups (PGs) are commonly used to prevent unwanted reactions and ensure that reactions occur only at specific sites on the sugar molecule[145,146]. The most common PGs are ester and ether groups, which can handle the acidic glycosylation conditions well. Often, auxiliaries[147–151] are used to control stereoselectivity in sugar synthesis, specifically to achieve a high stereoselectivity (Fig. 8b). In 2018, Rijs, Boltje, and their team reported the use of collision-induced dissociation

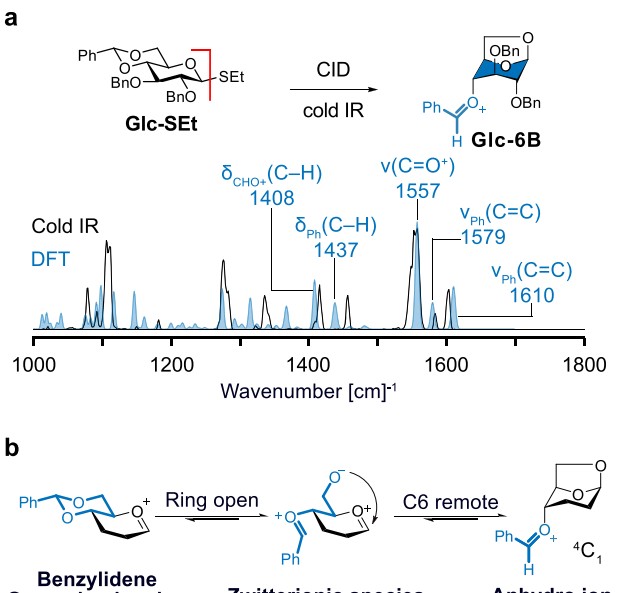

**Fig. 7 | IR analysis of a glucosyl cation with benzylidene groups. a** Experimental cryogenic (black line) and computed (blue) IR spectrum of 1,6-anhydro glucosyl cation carrying a benzylidene protecting group. δ are bending vibrations; *v* are stretching vibrations. Exemplary comparison of the cryogenic IR of a cation derived from a glucosyl donor by CID with the DFT calculated spectrum of a proposed structure. **b** Mechanism to account formation of anhydro cation. The formation of an anhydro cation with a unique PhCHO⁺ stretching was first discovered by cryogenic IR. Adapted from https://doi.org/10.1038/s44160-024-00619-0[58], under the terms of the CC BY 4.0 license, https://creativecommons.org/licenses/by/4.0/.

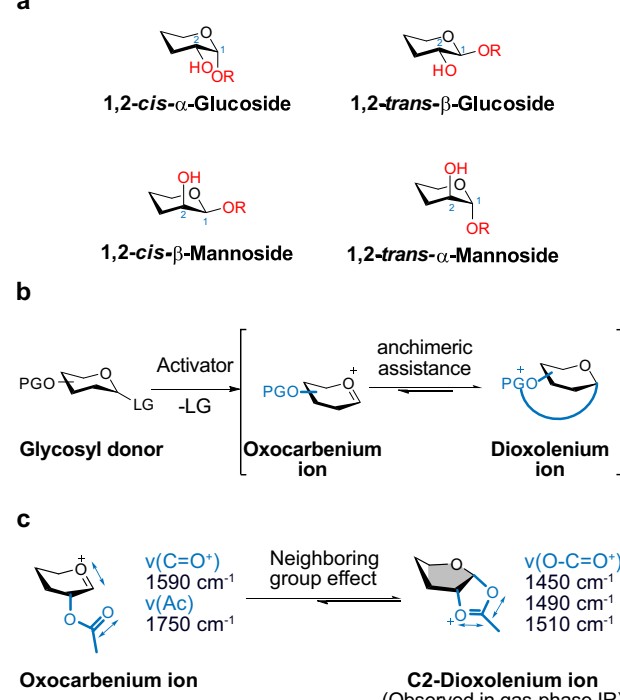

**Fig. 8 | A comprehensive study on glycosyl cations.** The intermediate rearrangement via anchimeric assistance is validated by gas-phase IR. **a** Types of glycosidic linkages. **b** The glycosylation reaction uses auxiliaries for intermediate stabilization and controlling of stereoselectivity. **c** Neighboring group effect via C2-acyl group participation. The IR spectra are distinct between oxocarbenium ions and C2-dioxolenium ions. R: substituent.

tandem mass spectrometry to produce glycosyl cations in the gas phase[60]. The resulting infrared spectra from infrared ion spectroscopy enabled detailed structural insights into these elusive glycosyl cations. Soon after, cryogenic IR spectroscopy was also used to study glycosyl cations[54]. The obtained IR data confirmed that a particular acyl group interaction, called "neighboring group participation", often plays a role in building 1,2-*trans* sugar linkages (Fig. 8c). The data agrees with the initial formation of an oxocarbenium ion, which is also formed in the CID process. Next, the C2-acyl group reacts with the anomeric carbon and forms a covalent bond on the 1,2-*cis* phase. The resulting C2-dioxolenium ion blocks the 1,2-*cis* side, promoting the formation of 1,2-*trans* glycosides. The IR spectra of glycosyl cations showed distinct chemical features that aided the structural identification by comparison with computed spectra. The peaks in the 1200 to 1800 cm$^{-1}$ range are diagnostic and correspond to vibrations of functional groups. For the oxocarbenium ion, there was a $\nu(\mathrm{C=O^+})$ stretch at 1590 cm$^{-1}$ and an acetyl group stretch at 1750 cm$^{-1}$. For the C2-dioxolenium ion, O-C-O stretches were observed at about 1450, 1490, and 1510 cm$^{-1}$. The C2-dioxolenium ion is commonly detected in gas-phase IR experiments due to the low free energy and high stability. The mechanism of the neighboring group effect was further refined and extended to a set of galactose derivatives[56,152].

The selective formation of 1,2-trans glycosidic linkages is most reliably achieved when a C2-acyl neighboring group is employed (Fig. 9). In particular, the C2-acyl group interacts with the anomeric carbon and forms a covalent bond. The existence of a C2-dioxolenium ion has been confirmed using IRMPD technology[60]. In this experiment, thiomannoside ManSPh was ionized and transferred into the gas phase. After collision-induced dissociation, a C2-dioxolenium ion is formed, and the vibration spectrum is recorded using IRMPD. The experimental IR spectrum (black line) matched closely with the DFT-simulated spectrum (purple filled). The most diagnostic peaks were found in the range of 1725–1800 cm$^{-1}$. These signals correspond to the C=O stretching of the acyl group. Bands at 1538 and 1496 cm$^{-1}$ correspond to the vibrations of the O-C=O$^+$ and C-C bonds.

1,2-*Trans* glycosides are generated by exploiting neighboring group participation and, from a synthesis point of view, are generally regarded easier to make than 1,2-*cis* glycosides[153]. A common, yet not fully understood, strategy to obtain 1,2-*cis* linkage involves long-range participation[154] (Fig. 10a). This strategy employs acyl PGs located at non-C2 positions of the sugar. During the glycosylation reaction, the acyl groups stabilize glycosyl cation and block the 1,2-*trans* face, promoting the opposite 1,2-*cis* nucleophilic addition. Cryogenic IR spectroscopy was applied to trap this key C4-dioxolenium ion in the gas phase[57]. The obtained data showed that the C4-Ac group contributes to this remote effect, but for the C6-Ac group the dioxolenium species was not observed. Recently, remote participation was studied in more detail by adjusting the electron density of the acyl group[155]. The acyl participation aligned with the armed-disarmed concept by Fraser-Reid[156,157], Ley[158] and Wong[159], where remote participation follows order of electron-rich pivaloyl (Piv) group > Acetyl (Ac) > electron-poor 4-trifluorated acyl (TFA) group. However, in chemical synthesis, α-glycosylation leads to a slightly different order, due to the potential influences of counter ion and solvent[41]. Ultimately, they provided an index of remote participation effectiveness in the order: 4-Piv ≥ 4-TFA > 4,6-di-Piv > 4-Ac > 6-Bn > 6-Ac.

Systematic studies on remote participation of acyl PGs have been reported by Codée and Boltje (Fig. 10b)[98]. For glucose, galactose, and mannose, remote participation at C3 and C4 is strong, while ring-opening side reactions occur with C6-Ac. The formation of a gas-phase intermediate is further linked to the stereoselectivity observed in condensed-phase glycosylation reactions. The gas phase studies on remote participation have been extended to uronic acids[160]. Mechanistically, the carboxylic ester forms

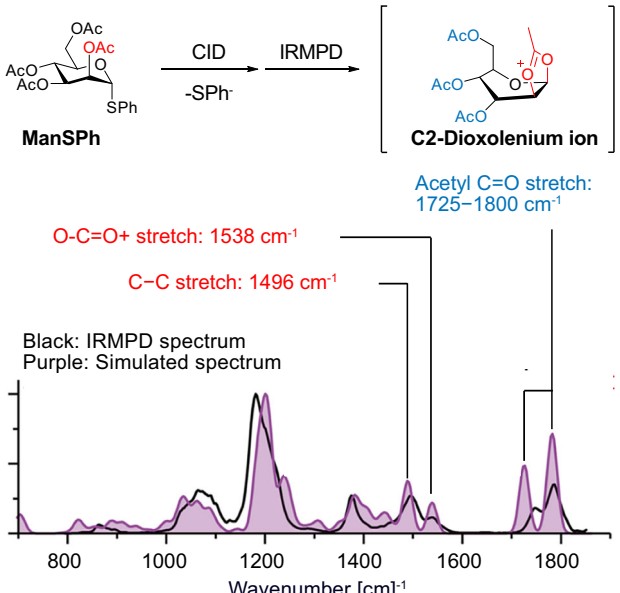

**Fig. 9 | The IRMPD spectrum of the mannosyl cation reported by Rijs and Boltje[60].** The IRMPD data (black line) and the calculated spectra (purple-filled) evidence the existence of a C2-dioxolenium ion. This structure data supports the C2-acyl neighboring group participation. Adapted from https://doi.org/10.1021/jacs.8b01236[60], under the terms of the CC BY 4.0 license, https://creativecommons.org/licenses/by/4.0/.

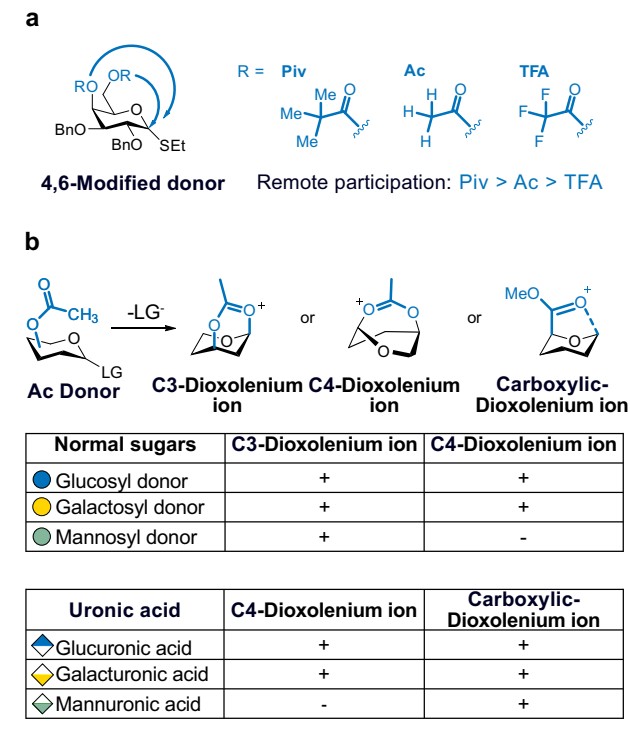

**Fig. 10 | The formation of the dioxolenium ion is identified through gas-phase IR spectroscopy.** The intermediate structure supports acyl participation in glycosylation reaction. **a** Remote participation. Participation of the C4-acyl group is confirmed, with the cyclization level controlled by the electron density of the acyl group. **b** Systematic probe on remote participation using a set of C3-Ac, C4-Ac, and C5-carboxylic ester donors. +: The dioxolenium ions are identified. −: not observed. **c** C4-Anchimeric assistance on Ferrier glycosyl cations. LG leaving group.

C5-dioxolenium ions. Alternatively, the axial C-4 acetyl group facilitates C4-dioxolenium ions. The participation among these two groups is competitive and dynamic. The intermediate ratio can be further defined by using isomer population analysis and conformational energy landscapes (CEL) research[44,161].

Ferrier rearrangement is developed to transform glycal into 2,3-unsaturated glycosyl derivatives (Fig. 10c). Several reviews on the Ferrier rearrangement provide an in-depth overview of this transformation[162]. In contrast to a typical glycosyl cation, the Ferrier rearrangement proceeds through allylic oxocarbenium ion, stabilized by conjugation into the double bond under acid conditions. This conjugation system directly influences the stereoselectivity of the glycosidic bond formation with an acceptor. Cryogenic IR spectroscopy is applied to probe the structure of the Ferrier cation, revealing a C4-acyl group interaction that leads to a C4-dioxolenium ion[55,139]. Besides, structural data evidence the anchimeric assistance of neighboring group participation at C4, supporting the development of building blocks for synthesizing 2,3-unsaturated glycosides.

Gas-phase IR spectroscopy on glycosyl cations has led to remarkable advancements in the understanding of key synthetic concepts in carbohydrate synthesis. To analyze the structure of glycosyl cations, several parameters in the DFT methods including scaling factors, and temperature must be optimized based on the instrumental setup. For cryogenic IR spectra, the temperature is typically set to 90 K, matching the conditions of a cold ion trap, while for IRMPD, the temperature is set to 298 K. Examples of the computational parameters chosen in glycosyl cation research are summarized in Table S1, including those used in investigations of neighboring group participation[54,56,60], remote participation[57,98,138], benzyl participation[152,163], Ferrier rearrangement[55,139], and the ring-restricted 4,6-O-benzylidene system[58,164]. The structural data obtained in the gas phase helped to clarify how auxiliaries influence the structure of sugar intermediates in controlling $S_N1$-glycosylation reactions.

## Correlation of gas-phase and solution structure

Mass spectrometric techniques open possibilities to gain structural information on charged reactive intermediates. The main concern is that the structural evidence obtained in gas-phase experiments might be biased by the experimental conditions. The correlation between gas-phase and condensed-phase structures has been intensely debated[165]. Current research on β-Galactosidase has shown that transfer to the gas phase resulted in proteins where the structure closely resemble native structures obtained from crystallography[166]. Similarly, the research on native protein structures indicated that, for low charge states, the condensed-phase structure is at least partially preserved after transfer to the gas phase[167].

From a physical chemistry perspective (Fig. 11a), gas-phase structures (with a relative permittivity $\varepsilon_r = 1$) are in a significantly different environment than those in aqueous solutions ($\varepsilon_r = 80$). However, typical glycosylation reactions occur in highly apolar solvents such as toluene ($\varepsilon_r = 2.4$) or dichloromethane ($\varepsilon_r = 8.9$), which have permittivity values much closer to vacuum ($\varepsilon_r = 1$). Recent results from computational calculations support this trend (Fig. 11b)[58]. DFT structures of 1,6-anhydro glycosyl cation were nearly identical irrespective of whether they were probed in solution or in the gas phase. The ring puckers remain the same as those adopted in the gas phase, although flexible side chains may be oriented slightly differently. Further it should be considered that molecules in the gas phase are widely spaced apart, or in other words almost infinitely diluted. Experimentally determined stereoselectivities of glycosylation reactions at highly dilute conditions support this correlation and align well with the structures observed in the gas phase[155].

The group of Codée investigated a potential reaction pathway for glycosylation using structural data from gas-phase experiments (Fig. 11c)[164]. They initially found C3-dioxolenium ions in gas-phase infrared-ion spectroscopy experiment. After that, they examined the structures surrounding the transition states by using DFT calculations to confirm their findings. The DFT data showed that the C3-dioxolenium ion forms from an α-glycosyl triflate through an intermediate conversion involving remote participation.

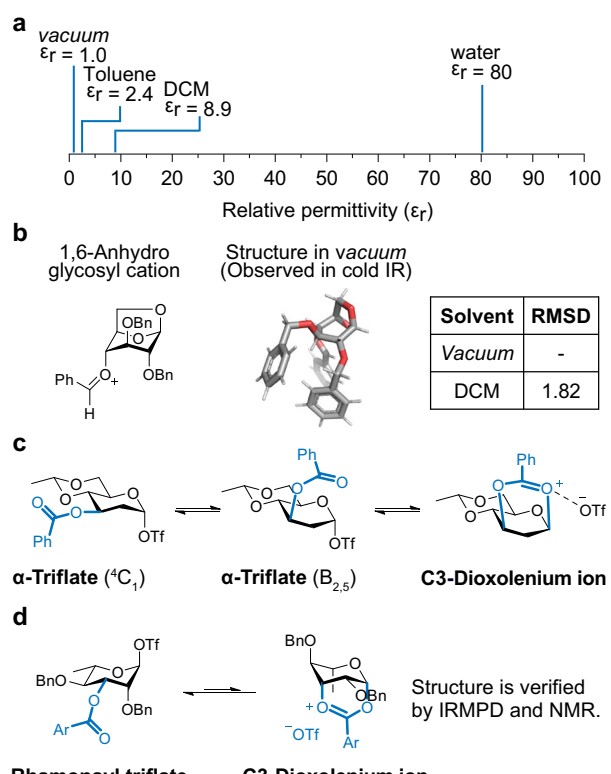

**Fig. 11 | Correlation between gas-phase glycosyl cations and solution synthesis. a** Relative electric permittivity: The vacuum structure resembles a highly apolar solvent structure, but it is significantly different from the structure of water. **b** The structure of an anhydro cation in dichloromethane is similar to the structure in a *vacuum*. **c** Transition state calculations to support the formation of the gas-phase C3-dioxolenium ion. **d** The presence of the C3-dioxolenium ion is confirmed in gas-phase IRMPD and solution by NMR. RMSD Root mean square deviation, DCM dichloromethane.

A key part of this process is the flipping of the sugar ring. When the glycosyl triflate changes shape from a chair ($^1C_4$) to a boat ($B_{2,5}$), the C3-acyl group moves to the upper side of the anomeric carbon. This change in shape helps the C3-acyl group interact to the anomeric carbon and allows the triflate (OTf) to leave. The resulting C3-dioxolenium ion is stabilized by the OTf anion, with a bond length of approx. 2.3 Å.

Boltje et al. applied a two-pronged strategy to validate the formation of ionic-type intermediates in glycosylation reactions (Fig. 11d)[40]. The kinetic profile suggests that α-rhamnosyl triflate is thermodynamic stable and can quickly equilibrium into the C3-dioxolenium ion through the participation of the C3-Ac group. This C3 cyclization does not require additional free counterions (OTf) from surrounding media. Besides, the C3-dioxolenium ion is high in energy and serves as a central species to control stereoselectivity under standard glycosylation conditions. Infrared-ion spectroscopy experiment confirmed the formation of a C3-dioxolenium ion in the gas phase. A consistent structure was observed in $CD_2Cl_2$ using EXSY and CEST NMR, where the glycosyl cations were trapped by a triflate (OTf) counter anion. $^1H$-$^{13}C$ HMBC NMR showed a correlation between the anomeric carbon and the quaternary carbon on the acyl-dioxolenium moiety.

## Outlook

This perspective highlights the recent achievements of MS-based technologies to study reactive sugar intermediates – glycosyl cations. The use of orthogonal techniques like IR spectroscopy and IMS has significantly enhanced our understanding of glycoside and glycosyl ion structures. Significant developments in instrumentation and laser technology in recent

years are starting to enable access to gas-phase spectrometry without highly specialized home-built setups. Likewise, dedicated computational strategies to interpret complex infrared spectra have been established. Through these computational calculations, the structures of many reactive intermediates have been discerned, leading to a detailed understanding of how auxiliary groups influence glycosidic bond formation. Although these approaches are limited to studying isolated systems due to the gas-phase conditions, many researchers have worked to bridge the gap between gas-phase data and solution data. A strong correlation between these two extreme conditions has been highlighted.

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

## Acknowledgements
K.P. acknowledges generous funding by the European Research Council, ERC-2019-CoG-863934-GlycoSpec. G.R.D.P. gratefully acknowledges the sponsorship of the Alexander von Humboldt Foundation. E.M. acknowledges generous funding by the Deutsche Forschungsgemeinschaft (DFG) under the GRK 2662-Project number 434130070. N.Ö. gratefully acknowledges the sponsorship of the Wenner-Gren Foundations. K.P., M.S., and G.M.V. acknowledge generous funding by the Deutsche Forschungsgemeinschaft (DFG, German Research Foundation) under the CRC 1449– Project ID 431232613.

## Author contributions
K.P. conceptualized this perspective. C.-W.C. summarized the history of carbohydrate chemistry and wrote the section on mechanistic studies on glycosylation reactions. D.W. organized the literature and provided a brief overview of computational studies. G.R.D.P. wrote the section on the infrared spectroscopy theory and introduced the design of IR instruments. E.M. covered the capabilities of ion mobility mass spectrometry. M.S. refined the section on infrared spectroscopy theory. L.B. and N.Ö. improved the portion on ion mobility mass spectrometry. G.V. polished the section on mechanistic studies of glycosylation reactions. In addition, all authors actively contributed to the writing and revision of the manuscript.

## Funding

## Competing interests
The authors declare no competing interests.
