## [Peer review file · Communications Chemistry]

Elucidating Reactive Sugar-Intermediates by Mass Spectrometry

Corresponding Author: Professor Kevin Pagel

Version 0:

Reviewer comments:

Reviewer #1

(Remarks to the Author)

The main purpose of the review by Kevin Pagel et al. is to provide an overview of the role played by two mass-spectrometry-based techniques, i.e. IRMPD action spectroscopy and IM-MS, to elucidate the structure of glycosyl cations, important and elusive intermediates in glycosylation reaction mechanisms.

The first part of the manuscript is focused on explaining the two different techniques; it is well written and can be useful for those who are not familiar with these methodologies. Perhaps the IMS section is too long, considering that only one result obtained with IMS is presented and could be shortened.

I make this suggestion because the second part, in particular the section called "Analysis of Glycosyl Cations using Gas-Phase IR Spectroscopy" which presents the results obtained with IRMPD action spectroscopy appears instead too concise and sometimes confusing while it deserves further clarification and discussion.

For example: 1) add a scheme to explain the concept of 1,2-trans or 1,2-cis-sugar linkages.

2) When the "neighboring group participation" results of figure 8b are described, it is unclear from the text whether the IRMPD spectrum agrees with the presence of both the "oxocarbenium" and "C2-bridged" ionic structures together in the sampled population or if the isomers were isolated and characterized separately.

3) in the paragraph "Ferrier rearrangement" the indication figure 8 e is missed.

4) In general, in the examples provided, it is not always clear what is the IRMPD contribution. Perhaps it is worth giving one less example but focusing more on describing the ionic intermediates that have been successfully characterized IRMPD action spectroscopy

Figure 8, which illustrates all the results reported in the aforementioned section also appears very confusing. It should be divided into several figures or if it is not possible, added some other figure in the SI to better explained the concept reported in figure 8.

In general, I recommend looking at the editing of the figures, often the writings are partially covered by the background of the figure and appear cut off.

Reviewer #2

(Remarks to the Author)

The authors present a perspective on the applications of diverse experimental techniques to assess the structures of the glycosyl oxocarbenium intermediates. These are extremely reactive chemical species whose isolation and experimental characterisation remained elusive until last decade. The authors highlight the use of IR spectroscopy and MS, assisted by computational methods to properly interpret the IR results. The perspective is excellent and provide a clear vision of the field, complementary to that provided by NMR (ref 13), which is advancing in a spectacular manner. The perspective is written in a precise and concise manner, with clear examples and highlighting the difficulties in accessing to these species and the ways to overcome these problems. It will be well received by the scientific community. The authors may also wish to cite a recent example by NMR, in which Thibaudeau and coworkers have advanced the super acid methodology to capture the naked cations without protecting groups: *J Am Chem Soc.* 2024 Nov 27;146(47):32618-32626. doi: 10.1021/jacs.4c11677. In any case, the perspective is excellent.

Reviewer #3

(Remarks to the Author)

The glycosyl cation is the cornerstone of glycosylation reaction but remains relatively poorly described as compared to other chemistries, which contributes to impede the full development of glycosciences. In this context, the authors review the advances on the understanding of the behaviour of the glycosyl cation in a very well written manuscript, supported by

abundant and relevant literature.

The manuscript starts with a brief historical perspective, followed by a focus on recent advanced MS-based gas phase approaches to produce, detect and characterise glycosyl cations.

The manuscript offers a well written overview to Ion mobilities technologies and vibrational ion spectroscopy schemes, which is very accessible to a non specialist reader. This section also includes a short description of the relevant computational approaches used to interpret IMS and spectroscopic data in terms of glycosyl cation properties, such as molecular structure, energies and reactivity.

The next section presents a clear and synthetic overview of the applications of vibrational ion spectroscopy to the characterisation of the glycosyl cation. It opens on an important discussion on the correlation between liquid phase and gas phase structure, which highlight the impact of gas phase studies for synthetic chemistry applications.

In conclusion, this review offers a very accessible introduction to the gas phase methodologies applied to the structural characterisation of the glycosyl cation, and how they are (somewhat counter-intuitively) relevant to the field of wet chemistry. I believe this work will contribute to establishing a common interdisciplinary ground for Physical Chemists and Organic Chemists, and will be of great interest to the readership of Communications Chemistry.

I fully support the publication of this manuscript with only the few remarks below.

Sincerely

- The introduction could benefit from a formal definition of the glycosyl cation and of some terminology (glycosyl, glucose, mannosyl , ...).
- Typical lifetimes in solution and in the gas phase could also be specified since they support a strong argument in favour of the gas phase angle.

Version 1:

Reviewer comments:

Reviewer #1

(Remarks to the Author)

I am very satisfied with the clear point-by-point letter from the authors and the changes made to the revised version. The manuscript is improved and can be published as is

Reviewer #1 (Remarks to the Author):

1) Comment: The main purpose of the review by Kevin Pagel et al. is to provide an overview of the role played by two mass-spectrometry-based techniques, i.e. IRMPD action spectroscopy and IM-MS, to elucidate the structure of glycosyl cations, important and elusive intermediates in glycosylation reaction mechanisms. The first part of the manuscript is focused on explaining the two different techniques; it is well written and can be useful for those who are not familiar with these methodologies. Perhaps the IMS section is too long, considering that only one result obtained with IMS is presented and could be shortened. I make this suggestion because the second part, in particular the section called “Analysis of Glycosyl Cations using Gas-Phase IR Spectroscopy” which presents the results obtained with IRMPD action spectroscopy appears instead too concise and sometimes confusing while it deserves further clarification and discussion. For example: 1) add a scheme to explain the concept of 1,2-trans or 1,2 -cis-sugar linkages.

We thank reviewer #1 for the suggestion. The concept of 1,2-trans and 1,2-cis glycosyl linkages has been addressed in the section “Analysis of Glycosyl Cations using Gas-Phase IR Spectroscopy.” Specifically, on page 7, left column, second paragraph, we state:

“The three-dimensional shape of carbohydrates directly influences their function in living organisms. The structure of a carbohydrate can for example differ in the configuration at the anomeric carbon (C1) where the orientation of the glycosidic bond leads to two possible forms, 1,2-cis glycosides and 1,2-trans glycosides (Figure 8a). Both anomers are important and commonly found in nature. The distinction between the two anomers depends on the relative positions of the C1-glycosidic bond and the C2-hydroxyl group in the sugar ring. For example, 1,2-trans glycosides are formed when the C1 group and the C2 group are on opposite sides. 1,2-cis glycosides, on the other hand, are formed when the C1 group and the C2 group are on the same side. Under this definition, β -glucoside is a 1,2-trans glycoside, and α -glucoside is a 1,2-cis glycoside. In mannose, the C2-hydroxyl group is in an axial position, which changes the relative orientation. As a result, α -Mannoside contains a 1,2-trans linkage while β -mannoside is 1,2-cis linkage.”

2) Comment: When the “neighboring group participation” results of figure 8b are described, it is unclear from the text whether the IRMPD spectrum agrees with the presence of both the “oxocarbenium” and “C2-bridged” ionic structures together in the sampled population or if the isomers were isolated and characterized separately.

We thank reviewer #1 for pointing out this concern. To clarify, we have provided additional information on the glycosyl cation by adding the following sentence: “The C2-bridged ion is commonly detected in gas-phase IR experiments due to the low free energy and high stability.” This has been included on page 8, left column, first paragraph, lines 8-10. Additionally, we have revised Figure 8c by adding the annotation “Observed in gas-phase IR” under the C2-bridged ion for further clarification.

3) Comment: in the paragraph “Ferrier rearrangement” the indication figure 8 e is missed.

Thank you. We have added the indication and referenced (Figure 10c) on page 9, left column, second paragraph, line 2. The figure number has been updated from Figure 8 to Figure 10 due to the addition of a new figure in the manuscript.

4) Comment: In general, in the examples provided, it is not always clear what is the IRMPD contribution. Perhaps it is worth giving one less example but focusing more on describing the ionic intermediates that have been successfully characterized IRMPD action spectroscopy

Answer: We thank reviewer #1 for this constructive suggestion. In response, we have provided a

focused example of the glycosyl cation by adding an IRMPD spectrum in Figure 9. Additionally, a more detailed explanation of the IR spectrum and the intermediate structure has been included on page 8, left column, second paragraph.

We now state:

“The selective formation of 1,2-trans glycosidic linkages is most reliably achieved when a C2-acyl neighboring group is employed (Figure 9). In particular, the C2-acyl group interacts with the anomeric carbon and forms a covalent bond. The existence of a 1,2-dioxonium ion has been confirmed using IRMPD technology.⁶⁰ In this experiment, thiomannoside ManSP_h was ionized and transferred into gas phase. After collision-induced dissociation, a 1,2-dioxonium ion is formed, and the vibration spectrum is recorded using IRMPD. The experimental IR spectrum (black line) matched closely with the DFT-simulated spectrum (purple filled). The most diagnostic peaks were found in the range of 1725–1800 cm⁻¹. These signals correspond to the C=O stretching of the acyl group. Bands at 1538 and 1496 cm⁻¹ correspond to the vibrations of the O-C=O⁺ and C-C bonds.”

5) Comment: Figure 8, which illustrates all the results reported in the aforementioned section also appears very confusing. It should be divided into several figures or if it is not possible, added some other figure in the SI to better explained the concept reported in figure 8. In general, I recommend looking at the editing of the figures, often the writings are partially covered by the background of the figure and appear cut off.

Answer: We thank reviewer #1 for the suggestion. In response, we have separated Figure 8 into Figures 8, 9 and 10. Additionally, the corresponding indication has been revised to reflect these changes.

Reviewer #2 (Remarks to the Author):

6) Comment: The authors present a perspective on the applications of diverse experimental techniques to assess the structures of the glycosyl oxocarbenium intermediates. These are extremely reactive chemical species whose isolation and experimental characterisation remained elusive until last decade. The authors highlight the use of IR spectroscopy and MS, assisted by computational methods to properly interpret the IR results. The perspective is excellent and provide a clear vision of the field, complementary to that provided by NMR (ref 13), which is advancing in a spectacular manner. The perspective is written in a precise and concise manner, with clear examples and highlighting the difficulties in accessing to these species and the ways to overcome these problems. It will be well received by the scientific community. The authors may also wish to cite a recent example by NMR, in which Thibaudeau and coworkers have advanced the super acid methodology to capture the naked cations without protecting groups: *J Am Chem Soc.* 2024 Nov 27;146(47):32618-32626. doi: 10.1021/jacs.4c11677. In any case, the perspective is excellent.

We thank reviewer #2 for highlighting this article. In the revised manuscript, we have incorporated the relevant work (ref: 53) and provided a brief introduction on page 2, right column, in the end of second paragraph. We now state: “Recently, Thibaudeau and coworkers have refined the superacid methodology to capture naked sugar cations without protective groups, offering deeper insights into the transition state of enzymatic reactions.⁵³”

Reviewer #3 (Remarks to the Author):

7) Comment: The glycosyl cation is the cornerstone of glycosylation reaction but remains relatively poorly described as compared to other chemistries, which contributes to impede the full development of glycosciences. In this context, the authors review the advances on the understanding of the behaviour of the glycosyl cation in a very well written manuscript, supported

by abundant and relevant literature. The manuscript starts with a brief historical perspective, followed by a focus on recent advanced MS-based gas phase approaches to produce, detect and characterise glycosyl cations. The manuscript offers a well written overview to Ion mobilities technologies and vibrational ion spectroscopy schemes, which is very accessible to a non specialist reader. This section also includes a short description of the relevant computational approaches used to interpret IMS and spectroscopic data in terms of glycosyl cation properties, such as molecular structure, energies and reactivity. The next section presents a clear and synthetic overview of the applications of vibrational ion spectroscopy to the characterisation of the glycosyl cation. It opens on an important discussion on the correlation between liquid phase and gas phase structure, which highlight the impact of gas phase studies for synthetic chemistry applications. In conclusion, this review offers a very accessible introduction to the gas phase methodologies applied to the structural characterisation of the glycosyl cation, and how they are (somewhat counter-intuitively) relevant to the field of wet chemistry. I believe this work will contribute to establishing a common interdisciplinary ground for Physical Chemists and Organic Chemists, and will be of great interest to the readership of Communications Chemistry. I fully support the publication of this manuscript with only the few remarks below. Sincerely

We thank reviewer #3 for the positive assessment.

8) Comment: - The introduction could benefit from a formal definition of the glycosyl cation and of some terminology (glycosyl, glucose, mannosyl , ...).

We thank reviewer #3 for highlighting this issue. In response, we have added the definition of glycosyl cation in the introduction on page 1, left column, third and fourth paragraph. We now state: “Gaining insight into the glycosylation processes is essential for controlling glycoside formation. Particularly, the outcome of glycosylation is largely determined by a highly reactive sugar intermediate — the glycosyl cation. This intermediate structure plays a key role in guiding the nucleophile approach. This, in turn, shapes the α - or β -configuration of the resulting glycosidic bond. The goal of mechanistic studies is to understand the behavior of glycosyl cations. This involves presenting a clear and step-by-step picture of how the intermediate participates in the reaction. However, the exact structure of glycosyl cations has remained unclear thus far. This lack of fundamental understanding is largely a result of their extremely short-lived nature. Their lifetimes are estimated to be just a few picoseconds in solution.^{13,14} In addition, glycosylation is a complex chemical process. It is influenced by various factors, including the solvent and acids. These large number of interconnecting factors directly impact the structure and stability of the intermediate. Due to these challenges, it is difficult to analyze glycosyl cations using conventional spectroscopic techniques.¹⁵”

Furthermore, on page 2, right column, lines 2-6, we now explicitly define glycosyl cations: “According to current understanding,^{35,36} glycosyl cations are positively charged molecules that form under acidic conditions. They originate from glycosyl donors when activated by an acid-based catalyst. Once formed, these cations are surrounded by counter-ions and solvent molecules.”.

9) Comment: - Typical lifetimes in solution and in the gas phase could also be specified since they support a strong argument in favour of the gas phase angle.

We thank reviewer #3 for highlighting the importance of the lifetime of glycosyl cations in spectrometric characterization. To address this, we have added the following statements:

“The goal of mechanistic studies is to understand the behavior of glycosyl cations. This involves presenting a clear and step-by-step picture of how the intermediate participates in the reaction. However, the exact structure of glycosyl cations has remained unclear thus far. This lack of

fundamental understanding is largely a result of their extremely short-lived nature. Their lifetimes are estimated to be just a few picoseconds in solution.^{13,14} In addition, glycosylation is a complex chemical process. It is influenced by various factors, including the solvent and acids. These large number of interconnecting factors directly impact the structure and stability of the intermediate. Due to these challenges, it is difficult to analyze glycosyl cations using conventional spectroscopic techniques.¹⁵ (Page 1, right column, second paragraph)
Additionally, we have included: “for seconds” (Page 1, right column, third paragraph, line 4).